# Cannabis use, mental health, and problematic Internet use in Quebec: A study protocol

**Magaly Brodeur**[1]*, **Didier Jutras-Aswad**[2,3], **Andrée-Anne Légaré**[1], **Adèle Morvannou**[4], **Eva Monson**[4], **Julie-Christine Cotton**[4], **Anders Hakansson**[5], **Virginie Parent**[6], **Catherine Hudon**[1]

1 Department of Family Medicine and Emergency Medicine, Université de Sherbrooke, Sherbrooke, Quebec, Canada, 2 Department of Psychiatry and Addiction, Université de Montréal, Montreal, Quebec, Canada, 3 Research Centre, Centre Hospitalier de l'Université de Montréal, Montreal, Quebec, Canada, 4 Department of Community Health Sciences, Université de Sherbrooke, Sherbrooke, Quebec, Canada, 5 Faculty of Medicine, Clinical Addiction Research Unit, Lund University, Lund, Sweden, 6 Centre Integré Universitaire de Santé et de Services Sociaux de l'Estrie, Centre Hospitalier Universitaire de Sherbrooke, Sherbrooke, Quebec, Canada

* Magaly.Brodeur@USherbrooke.ca

**Data Availability Statement:** No datasets were generated or analyzed during the current study. Upon study completion, the data generated for this

## Abstract

### Background

Problematic Internet use is characterized by excessive use of online platforms that can result in social isolation, family problems, psychological distress, and even suicide. Problematic Internet use has been associated with cannabis use disorder, however knowledge on the adult population remains limited. In Quebec, cannabis use has significatively increased since 2018, and it is associated with various risks in public safety, public health, and mental health. This study aims to identify factors associated with problematic Internet use among adult cannabis users and to better understand their experiences.

### Method

This project is a mixed explanatory sequential study consisting of two phases. Phase 1 (n = 1500) will be a cross-sectional correlational study using probability sampling to examine variables that predispose individuals to problematic Internet use, characteristics associated with cannabis use, Internet use, and the mental health profile of adult cannabis users in Quebec. Descriptive analyses and regression models will be used to determine the relationship between cannabis use and Internet use. Phase 2 (n = 45) will be a descriptive qualitative study in the form of semi-structured interviews aimed at better understanding the experience and background of cannabis users with probable problematic Internet use.

### Discussion

The results of this study will support the development of public policies and interventions for the targeted population, by formulating courses of action that contribute to the prevention and reduction of harms associated with cannabis use and problematic Internet use. Furthermore, an integrated knowledge mobilization plan will aid in the large-scale dissemination of

research will be available upon reasonable request to the corresponding author. Data will not be publicly available due to potentially identifying information that could compromise the privacy of research participants.

**Funding:** Magaly Brodeur (MB) received funding for this study by the Fonds de recherche du Québec - Santé (FRQS, https://frq.gouv.qc.ca/sante/), in partnership with the Ministère de la Santé et des Services sociaux (MSSS) and the Fonds de recherche du Québec - Société et culture (FRQSC) under the Research Program on the Non-Medical Use of Cannabis: Impacts of Consumption on Health (grant # 324497). The funders did not and will not have a role in study design, data collection and analysis, decision to publish, or preparation of the manuscript.

**Competing interests:** The authors have read the journal's policy and have the following competing interests: MB is a first-line physician and professor at Université de Sherbrooke. None of her research projects are funded by the gambling industry. DJA holds a clinical scientist career award from Fonds de Recherche du Québec (FRQS). DJA receives investigational products from Cardiol Therapeutics for a clinical trial funded by Quebec Ministry of Health and Social Services outside of the submitted work. There are no patents, products in development or marketed products associated with this research to declare. This does not alter our adherence to PLOS ONE policies on sharing data and materials.

information that can result useful to decision-makers, practitioners, members of the scientific community, and the general population regarding the use of cannabis and the Internet.

## Introduction

In Quebec, after alcohol and tobacco, cannabis is the most widely used psychoactive substance among the population. Since its legalization in 2018, cannabis use has been increasing in Quebec (14% in 2018, 16.4% in 2019, and 20% in 2021) [1,2]. Additionally, cannabis consumption is associated with numerous risks to public safety (e.g., impaired driving), public health (e.g., harms associated with cannabis use on the individual, loved ones, and society in general), and mental health [3]. Cannabis users, particularly those with a cannabis use disorder, are at high risk of developing a concurrent mental health disorder (e.g., anxiety disorder, depressive disorder, psychotic disorder), another substance use disorder (e.g., alcohol, cocaine), or a behavioral addiction [4,5].

Behavioral addictions result when the object of addiction is a behavior rather than a psychotropic product [6]. Problematic use of the Internet and new technologies (hereafter, problematic Internet use) is a behavioral addiction characterized by excessive use of online activities that can take many forms: cyber relationships and social media (e.g., Facebook, Instagram, Tinder); video games and hybrid games (e.g., Candy Crush); shopping (e.g., Amazon); video viewing (e.g., Netflix, Youtube); information searching (e.g., Google); etc. [6]. The consequences of problematic Internet use, such as social isolation, family problems, psychological distress, suicide, etc. are devastating for those affected and their loved ones [7,8]. The COVID-19 pandemic has made problematic Internet use a priority issue for many entities both in Quebec and abroad (government, researchers, health professionals, etc.), as it has led to increased use of the Internet and new technologies [9,10]. Additionally, studies conducted during the COVID-19 pandemic suggest associations between Internet-related addictions and various types of mental health problems, including anxiety, depression, stress, psychological distress, loneliness, cyberchondria, and sleep disorder [11].

In the literature, cannabis use, specifically cannabis use disorder, often appears to be associated with problematic Internet use [12–14]. Certain groups, such as adolescent cannabis users, have been identified as being at increased risk for problematic Internet use [12,15,16]. However, knowledge of the adult population remains limited [17]. Therefore, the relationship between cannabis use and problematic Internet use and its associated risks remains difficult to understand given the paucity of studies exploring the mechanisms underlying this association in the adult population [17]. Knowledge development about the risks associated with cannabis use and mental health, and more specifically, problematic Internet use, is essential to guide the steps taken by public entities in the field.

In line with the *Plan d'action interministériel en dépendances* of the Governement of Quebec (Interministerial Action Plan on Addictions, PAID) [18], this project adopts a harm reduction perspective regarding cannabis consumption and problematic Internet use. This approach is partly inspired by a renewed vision of the "Agent-Host-Environment" epidemiological triad [17]. Although this framework was developed for the field of epidemiology, it has been expanded and used as a frame of reference in the field of addictions: 1) the host refers to people who use cannabis and the Internet, and their biopsychosocial characteristics (including sex and gender); 2) the agent refers to cannabis and the Internet, and their intrinsic characteristics; and 3) the environment refers to the social, economic, political, and legislative context that

surrounds cannabis use and Internet use [19,20]. The advantage of this framework is its ability to explain the issues associated with cannabis use and problematic Internet use and the harms associated with them, using different perspectives [19].

## Objectives

This project aims to:

1. Identify factors associated with problematic Internet use among cannabis users, such as habits surrounding their cannabis consumption and Internet use, use of other psychoactive substances, factors related to mental health, etc.

2. Understand the experience and background of cannabis users with problematic Internet use.

## Materials and methods

This project will be conducted as a mixed explanatory sequential study [21] consisting of two phases that will be carried out over a span of three years [22].

### Phase 1: Cross-sectional correlational study

The first phase will constitute a self-reported online questionnaire (n = 1500) via a web panel, which will identify factors associated with problematic Internet use among cannabis users.

### Ethics statement

This research project has been ethically and scientifically approved by the Research Ethics Committee and Scientific Evaluation Committee of the *Centre intégré universitaire de santé et de services sociaux (CIUSSS) de l'Estrie—Centre hospitalier universitaire de Sherbrooke (CHUS)* on July 19, 2023 (reference number: 2024-5139-CyberD-Cannabis; see S1 and S2 Files). Written consent will be obtained from all participants in this study prior to their participation and participants will be informed of their right to withdraw at any time.

### Recruitment and sampling

To be eligible in the study, participants must have used cannabis at least once a week in the past 12 months, be 18 years of age or older, and reside in Quebec. Recruitment will employ probability sampling (i.e., stratified simple random) and will be done by email through a firm specializing in surveys and web panels. The target sample size is 1500 people. Recruitment through the web panels firm, which reaches over 200,000 members in Quebec, will be carried out to ensure that the Quebec population is representative in terms of age, sex and gender, language, and geographic region. This will be done by generating a sample representative of the general Quebec population (according to Statistics Canada) and asking filter questions based on the aforementioned variables to the firm's participant pool, ultimately identifying our target population. As compensation for their time, participants will receive credits redeemable for cash through the surveys and web panels firm's website.

### Sample size calculation

Sample size (n = 1500) was determined according to the number of participating individuals needed for the development of a regression model composed of 10 variables, i.e., 10

participants per variable [23], with enough statistical power (0.8) and able to detect a relatively small effect size (0.11) in the relationship between cannabis use and Internet use [24]. This sample size will also allow us to obtain a congruent sample size for phase 2 of the study (n = 45) based on participant measurements of Internet addiction and the estimated prevalence of problematic Internet use (6 to 15%) [14,25].

## Data collection

Data will be collected through a bilingual (French and English) online questionnaire. The questionnaire, composed of validated tools in English and French, will be divided into 5 sections: 1) Sociodemographic profile; 2) Cannabis and other psychoactive substance use profile; 3) Internet use profile; 4) Link between cannabis use and Internet use and impacts; 5) Mental health profile. A pre-test will be conducted with 30 people during the month preceding the launch of the questionnaire. At the end of the questionnaire, people will be asked to indicate their interest in participating in phase 2 of the study.

## Variables and measurement instruments

**Sociodemographic profile.** Questions for the measurement of sociodemographic variables will be taken from the Canadian Community Health Survey (CCHS) [26]: e.g., age, language, religion, marital status, education level, employment, income (personal and household), country of birth, immigration status and racial or cultural group. The sex and gender of participants will be requested to conduct a sex- and gender-specific analysis (see Sex and Gender Analysis section).

**Profile of cannabis and other psychoactive substance use.** Questions taken from the Quebec Cannabis Survey (QCS) will make it possible to draw a profile of cannabis use among participants (non-medical and/or medical use, age of first use of cannabis, method of use (joint, pipe, dabbing, vaping, etc.), form of the product (flower, hashish, extract or liquid concentrate, food product, drink), quantity consumed, frequency of consumption, etc.) as well as their use of other psychoactive substances [27]. Problematic cannabis use will be measured with the Cannabis Abuse Screening Test (CAST)[28]. The CAST is a 6-item questionnaire that assesses the level of risk associated with cannabis use. This questionnaire is commonly used in the context of research with the general population [29]. Each of the 6 items refers to a problematic situation encountered during cannabis use. The items are rated on a 5-point Likert scale. A score between 0 and 4 is established for each answer and the total score is obtained by adding the items (0–24): no risk of dependence ($<3$), low risk of dependence (3–6), high risk of dependence ($>8$) [28]. The use of other psychoactive substances will be assessed with the Alcohol, Smoking and Substance Involvement Screening Test (ASSIST) [30].

**Profile of Internet use.** Questions from the Canadian Internet Use Survey (CIUS) [31] will be used to develop a profile of Internet usage among participants (accessibility, types of devices held (e.g., smartphone, tablet, computer), time of use, frequency of use). Questions taken from the *Dépistage/Évaluation du Besoin d'Aide—Internet* (DÉBA-Internet) will allow us to draw a profile of Internet use (type of use (e.g., cyber-relations and social media, video games and hybrid games, gambling, shopping, video viewing, information search), etc.) [32]. Finally, problematic Internet use (our dependent variable) will be assessed with the Internet Addiction Test (IAT) [25]. The IAT consists of 20 questions using a Likert scale. A score between 0 and 5 is established for each answer and summed to give a total score. A score between 0 and 30 reflects normal Internet use; a score between 31 and 49 indicates a probable mild level of addiction; a score between 50 and 79, a probable moderate level and a score between 80 and 100, a probable severe addiction [25].

**Relationship between cannabis use and Internet use and impacts.** Questions to assess the impacts of cannabis use and Internet use (e.g., concurrent use, change in frequency and time of use) in different areas of participants' lives (work, relationships, school, etc.) will be adapted from the CQS [27]. Questions regarding the links between cannabis use and Internet use will be developed by the research team in collaboration with the project's steering committee (composed of stakeholders in the field who will be involved in all phases of the study).

**Mental health profile.** The presence of anxiety symptoms will be assessed using the Generalized Anxiety Disorder-7 (GAD-7), depressive symptoms with the Patient Health Questionnaire-9 (PHQ-9) [33,34]. A mental health and treatment inventory will determine the mental health profile of participants (history of mental and active disorders in the past 12 months (depression, anxiety, etc.), known 1st-degree family history, treatment, etc.). Questions will be taken from the Mental Health and Access to Care Survey (MHACS) [35].

The standardized tools used in this questionnaire have all been validated and are available in French and English [25,29,30,33,34,36–39].

## Data analysis

Descriptive analyses will be used to depict the characteristics of the individuals participating in the study and to create a profile of their cannabis and Internet use. In order to meet objective 1, multivariate logistic regression models will be used to identify variables that predispose to problematic Internet use (independent variables: e.g., age, sex, gender, marital status, employment, substance use (alcohol, drugs, etc.), characteristics associated with cannabis use (frequency, method of use, type of product, age of initiation, quantity consumed, etc.), characteristics of Internet use (type of use, time of use, frequency of use, etc.), PHQ-9 score (depressive symptoms), GAD-7 score (anxiety symptoms), etc.) as a function of IAT score (dependent variable—continuous type) [40,41]. Odds ratios will also be reported. Descriptive analyses and regression models will be performed with SPSS ® software.

## Phase 2: Descriptive qualitative study

In phase 2, semi-structured interviews will be used (n = 45) to better understand the experience and background of cannabis users with probable problematic Internet use [42,43].

## Recruitment and sampling

Inclusion criteria to participate in the semi-structured interviews are 1) have participated in Phase 1 of the study, 2) have agreed to be contacted for Phase 2, and 3) present a problematic use of the Internet, according to IAT score during Phase 1, i.e., a score greater than 50 [25]. Sampling will be targeted and with maximum variation [44] using the following criteria: age, sex, gender, and region of residence. This will allow for the study and understanding of the experience and background of cannabis users with problematic Internet use according to different profiles. Three groups of 15 individuals will be recruited (n = 45): 1) individuals with probable cannabis use disorder (CAST Score > 8); 2) individuals who use cannabis regularly (i.e., 1–6 days per week); 3) individuals who use cannabis daily (i.e., every day) without probable cannabis use disorder [27].

Recruitment will be conducted by members of the research team via email from the list of participating individuals who have agreed to be contacted for Phase 2 and meet the inclusion criteria. Individuals who do not respond to the invitation will be contacted again after 7 days by telephone. A gift card (e.g., grocery store, pharmacy) worth $50 will be given to each participant as compensation for their time.

## Data collection

Semi-structured interviews (60 to 90 minutes) will be conducted in French or English. They will be conducted in person or virtually, depending on the participant's preference, and the research team's availability. The interviews will be conducted by two team members with expertise in qualitative methods and experience in semi-structured interviews.

As expected in a sequential explanatory design, the results of phase 1 will help guide the sampling and the development of the semi-structured interview guide for phase 2 and identify areas requiring further investigation (e.g., links between cannabis use and Internet use and its impacts) [45]. The interview guide, which will be pre-tested on a sample of five individuals, will be structured to better understand the experience and background of cannabis users with probable problematic Internet use [44,46]. Open-ended questions will be used to explore the dynamics of use and to understand how cannabis use relates to their daily lives (interpersonal relationships, work, study, etc.) as well as to their overall mental health. The interviews will be recorded in digital audio format and then transcribed verbatim and anonymized.

## Data analysis

An inductive and deductive thematic analysis based on the "agent-host-environment" theoretical framework [19] will be carried out [44]. The transcription of the interviews will first be independently coded and classified by themes by two members of the research team. These themes will then be grouped into similar groups to create a thematic tree structure. The team will carry out iterative analysis phases to ensure a thorough understanding of the analysis content and coding process, as well as to establish a shared comprehension of the context and data being studied [21]. This iterative process will allow for the triangulation of team members' expertise (medical, psychological, etc.) and knowledge. Recruitment will be done following a data saturation grid and will be carried out until the data are saturated [47]. Once the iterative coding is complete, the team will present their findings and proposals, supported by qualitative data and emerging themes, to address the second objective of the project. NVivo ® software will be used for all qualitative data analysis.

**Differential analysis by sex and gender.**   This project recognizes the difference between the concepts of sex and gender. According to the Canadian Institutes of Health Research (CIHR), sex refers to a set of biological attributes in humans and animals related primarily to physical and physiological characteristics, such as chromosomes, gene expression, hormone levels, and the anatomy of the reproductive system. On the other hand, gender refers to the roles, behaviors, expressions, and identities that society constructs [48]. Sex and gender play a major role in addiction studies [49]. It has been evidenced that sex has an influence on the metabolism of psychoactive substances and their physiological effects [49], while gender influences the profiles, contexts, and dynamics of use and experience as well as effective approaches to prevention and intervention [50].

In this project, sex and gender considerations will be integrated into all phases of the project, from project inception to knowledge mobilization to project closure. In Phase 1 of the project, the self-reported questionnaire will obtain the sex and gender of participants. Descriptive data will be presented by sex and gender and multivariate models will be conducted for sex and gender. To achieve an inclusive approach, variables such as sexual orientation and ethnic group will be included in the analyses. In Phase 2 of the study, the sex and gender of participants will be part of the maximum variation criteria for recruitment. Data from the semi-structured interviews will be coded and analyzed by sex and gender to better understand the experience and background of cannabis users with problematic Internet use by sex and gender.

At the end of the study, we will provide evidence-based policy options that consider sex and gender. These courses of action will allow for the implementation of "specific actions adapted to each person's sex, gender [and] needs", a need identified in the PAID [18]. It is also important to note that the final study report and all knowledge mobilization materials (video vignettes, outreach texts, etc.) will present data by sex and gender, highlighting, among other things, sex and gender differences in cannabis use and problematic Internet use.

## Integration phase

An integration phase will be conducted after both phases of the project. Matrices, diagrams, graphs, and tables will be created to integrate quantitative and qualitative data [45,51].

A co-construction workshop will be carried out to formulate courses of action using the *Technique de Recherche d'Informations par Animation d'un Groupe Expert* (Technique of Information Retrieval by Animation of a Group of Experts, TRIAGE Method) [52,53]. This workshop, conducted with members of the project's steering committee, will provide valuable insights for the development of effective strategies to mitigate the negative impacts of cannabis and problematic Internet use. The TRIAGE method includes four stages: 1) preparation; 2) individual production; 3) compilation; and 4) collective production. The preparation phase will be carried out by members of the research team [53]. To support the deliberation of the participating individuals, the matrices, diagrams, graphs, and tables produced during the integration phase will be distributed in order to present the quantitative and qualitative data of the study in an integrated manner during the individual production phase [45]. During this phase, participants will be invited to formulate and reflect on priority courses of action aimed at preventing and reducing the harms associated with cannabis use and problematic Internet use, considering the results of the study. This information will then be transmitted to the members of the research team who will compile the responses in preparation for the collective production phase. This next phase will make it possible to select, by consensus, the courses of action to be prioritized according to the sex and gender of people who use cannabis and have problematic Internet use [53]. At the end of this workshop, key messages for all groups that will benefit from the project (people who use cannabis and the Internet, partners in the call for proposals, decision-makers, practice settings, the general public, scientists, etc.) will also be developed in collaboration with the steering committee.

## Discussion

### Anticipated benefits

This study has many strengths. To our knowledge, this is the first mixed-methods study of cannabis use and problematic Internet use in Quebec and internationally. It will identify factors associated with problematic Internet use among cannabis users and better understand the experience and background of cannabis users with problematic Internet use. The data collected will be essential for the development of effective public policies in line with the needs of the population studied and the stakeholders in the field. This study is also distinguished by its strong partnership component. It includes a steering committee made up of stakeholders in the field who will be involved in all phases of the study to ensure that the project is in line with the needs of the stakeholders and to achieve an integrated mobilization of knowledge throughout the project (see Knowledge Mobilization). Ultimately, this study will have considerable concrete benefits at different levels.

This will be one of the first studies at the international level to specifically address the risks associated with cannabis use and problematic Internet use in the general adult population. The mixed-methods design will make it possible to identify the factors associated with problematic

Internet use among cannabis users and to better understand the experience of people affected by this problem, which is fundamental to the development of interventions for this population.

In line with the guiding principles of the PAID, this study will support the development of public policies by formulating courses of action to contribute to the prevention and reduction of harm associated with cannabis use and problematic Internet use by informing, raising awareness, and supporting decision-makers and managers. Call for proposal meetings with partners, including the Quebec Ministry of Health and Social Services (MSSS) will ensure that the needs of the parties involved are met and that measures are identified to facilitate the implementation of measures resulting from the project.

This study will contribute to training the next generation of research students (at the undergraduate, master's, and doctorate levels). We will offer various possibilities (summer internships, part-time employment, thesis, etc.) to involve them in all phases of the project and familiarize them with this type of research project including a strong partnership component. Furthermore, the results of this project will educate the public about the harms associated with cannabis use and problematic Internet use (see Knowledge Mobilization section) and contribute to prevention and harm reduction in this area.

Ultimately, the project will have various effects. A few examples of how the results could be applied by the various stakeholders at the end of the project include awareness campaigns (traditional and social media) on the harms associated with cannabis use and Internet use in collaboration with the MSSS, implementation of online visual messages aimed at raising awareness among cannabis users (e.g., social networks, *Société Québécoise du cannabis* (SQDC) website, etc.), the adaptation of interventions measures based on sex and gender for people who use cannabis and have a problematic Internet use, etc. The net goal is for this study to be a first step towards the implementation of concrete measures aimed at preventing and reducing harms associated with problematic Internet use among cannabis users.

## Knowledge mobilization

Knowledge mobilization for this project will be achieved through an integrated co-construction action plan via the involvement of stakeholders sitting on the project steering committee. This plan will be oriented towards large-scale dissemination in Quebec, in Canada, and internationally, in collaboration with our partners. The knowledge mobilization plan will include different activities and aims to highlight and disseminate the results of the project to five (5) target populations: decision-makers, managers, and partners of the call for proposals; practitioners and practice settings; the general public (including people who use cannabis and the Internet as well as their relatives); the scientific community; and the student community. Knowledge mobilization activities include reports at the end of each phase of the study for stakeholders, an online training session centered around the themes of the study open to all students and workers of the labor market, dissemination of educational material adapted to the general public, participation in national and international research conferences and publication of results in scientific journals.

## Limitations

Conducting a study of this type involves certain challenges. The main challenge is the number of participants to be recruited. In aims to facilitate recruitment, a firm specializing in surveys and web panels will allow us to recruit a random sample of participants (N = 1500). Given that our study is interested in people who use cannabis and use the Internet, recruitment via a web panel (including only people with access to the Internet), is appropriate in this context and

poses fewer challenges in terms of representativeness. Participants invited to participate in Phase 2 of the study will be recruited from this pool. The second challenge is the collection of self-reported data during the cross-sectional portion of the study. By answering the questionnaire, participants could, for example, minimize certain habits or symptoms, i.e., a social desirability bias. To minimize this bias, we will mainly use validated scales during the quantitative phase of the study. A third challenge will be to carry out the qualitative phase. This phase will generate a significant amount of data. To do so, we will use various tools including NVivo software and benefit, as in phase 1 where SPSS software will be used, from the combined expertise of the team members. Finally, in aims to foster a caring environment for the project, all team members and members of the steering committee will attend a workshop on unconscious bias in research as well as equity, diversity, and inclusion.

## Supporting information

**S1 File. IRB/Research Ethics Committee approval letter (ORIGINAL).**
(PDF)

**S2 File. IRB/Research Ethics Committee approval letter (ENGLISH).**
(PDF)

## Acknowledgments

We would like to thank Annie Desjardins (patient-partner), Anne-Marie Auger (research coordinator) and Natalia Muñoz Gómez (research professional) for their involvement in the study and the revision of the protocol.

## Author Contributions

**Conceptualization:** Magaly Brodeur.

**Funding acquisition:** Magaly Brodeur, Didier Jutras-Aswad, Andrée-Anne Légaré, Adèle Morvannou, Eva Monson, Julie-Christine Cotton, Anders Hakansson, Virginie Parent, Catherine Hudon.

**Methodology:** Magaly Brodeur, Didier Jutras-Aswad, Andrée-Anne Légaré, Adèle Morvannou, Eva Monson, Julie-Christine Cotton, Anders Hakansson, Virginie Parent, Catherine Hudon.

**Project administration:** Magaly Brodeur.

**Writing – original draft:** Magaly Brodeur.

**Writing – review & editing:** Magaly Brodeur, Didier Jutras-Aswad, Catherine Hudon.

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
