## [Decision Letter · Decision Letter 0]

28 Feb 2024

PONE-D-23-22618

Cannabis use, mental health, and problematic Internet use in Quebec: A Study Protocol

PLOS ONE

Dear Dr. Brodeur,

Thank you for submitting your manuscript to PLOS ONE. After careful consideration, we feel that it has merit but does not fully meet PLOS ONE’s publication criteria as it currently stands. Therefore, we invite you to submit a revised version of the manuscript that addresses the points raised during the review process.

Reviewer 1 has some very important points that would improve the protocol and I encourage to submit another revision with these points addressed. 

We look forward to receiving your revised manuscript.

Kind regards,

Souparno Mitra, M.D.

Academic Editor

PLOS ONE

Journal Requirements:

   "I have read the journal's policy and the authors of this manuscript have the following competing interests: Magaly Brodeur (MB) is a first-line physician and professor at Université de Sherbrooke. None of her research projects are funded by the gambling industry. She works as a prevention and harm reduction consultant for Loto-Quebec, the crown corporation responsible for state-run gambling in the province of Quebec (Canada). 

Didier Jutras-Aswad (DJA) holds a clinical scientist career award from Fonds de Recherche du Québec (FRQS). DJA receives investigational products from Cardiol Therapeutics for a clinical trial funded by Quebec Ministry of Health and Social Services."

Additional Editor Comments:

Comments from PLOS Editorial Office: We note that one or more reviewers has recommended that you cite specific previously published works. As always, we recommend that you please review and evaluate the requested works to determine whether they are relevant and should be cited. It is not a requirement to cite these works. We appreciate your attention to this request.

Reviewers' comments:

Reviewer's Responses to Questions

**Comments to the Author**

1. Does the manuscript provide a valid rationale for the proposed study, with clearly identified and justified research questions?

Reviewer #1: Yes

Reviewer #2: Yes

2. Is the protocol technically sound and planned in a manner that will lead to a meaningful outcome and allow testing the stated hypotheses?

Reviewer #1: Partly

Reviewer #2: Yes

3. Is the methodology feasible and described in sufficient detail to allow the work to be replicable?

Reviewer #1: No

Reviewer #2: Yes

4. Have the authors described where all data underlying the findings will be made available when the study is complete?

Reviewer #1: No

Reviewer #2: Yes

5. Is the manuscript presented in an intelligible fashion and written in standard English?

Reviewer #1: Yes

Reviewer #2: Yes

6. Review Comments to the Author

You may also provide optional suggestions and comments to authors that they might find helpful in planning their study.

Reviewer #1: I believe that the protocol can be improved with the following comments.

1. ll74-75, the statement “The COVID-19 pandemic has made problematic Internet use a priority issue for many…” needs more citation in addition to the current consensus statement. The following references may work.

Alimoradi, Z., Lotfi, A., Lin, C.-Y., Griffiths, M. D., & Pakpour, A. H. (2022). Estimation of behavioral addiction prevalence during COVID-19 pandemic: A systematic review and meta-analysis. Current Addiction Reports, 9, 486-517.

Ruckwongpatr, K., Paratthakonkun, C., Ghavifekr, S., Gan, W. Y., Tung, S. E. H., Nurmala, I., Nadhiroh, S. R., Pramukti, I., & Lin, C.-Y. (2022). Problematic Internet Use (PIU) in Youth: A Brief Literature Review of Selected Topics. Current Opinion in Behavioral Sciences, 46, 101150.

Chen, I.-H., Chen, C.-Y., Liu, C.-h., Ahorsu, D. K., Griffiths, M. D., Chen, Y.-P., Kuo, Y.-J., Lin, C.-Y., Pakpour, A. H., Wang, S.-M. (2021). Internet addiction and psychological distress among Chinese schoolchildren before and during the COVID-19 outbreak: A latent class analysis. Journal of Behavioral Addictions, 10(3), 731-746.

2. Line 80, I think that (10,13,14) should be [10,13,14].

3. When the authors propose their first objective (i.e., identify factors associated with problematic Internet use among cannabis users), it is unclear to me if this objective is based on hypotheses or is exploratory. This is important because if the objective will be investigated using hypotheses, the authors should provide theoretical background to support. However, if it is exploratory, the authors do not need to provide theoretical background. But even it is exploratory, the authors should provide some directions regarding what factors are of interests.

4. The authors want to recruit 1500 participants for the first phase of the present project, and the proposed sample size calculation is vague. Specifically, the authors simply said that “Sample size (n = 1500) was determined according to the number of participating individuals needed for the development of a regression model, composed of 10 variables associated with the epidemiological triad "host-agent-environment" [17] to determine the relationship between cannabis use and Internet use”. How exactly the sample size is calculated remains unclear to me.

5. Following the previous comment, I think that “sample size calculation” deserves to be a subsection instead of mentioning in the Data analysis section.

6. It is unclear to me how the authors can recruit the participants to be representative in terms of age, sex and gender, language, and geographic region. Say, what are the proposed sample size for each stratified group?

7. Also, it is unclear how the representativeness of the database in the firm which will help distribute the questionnaire.

8. The authors said, “Problematic Internet use has been associated with cannabis use disorder, especially among adolescent users” in the Abstract. However, the present protocol excludes adolescents. Then, why do the authors want to emphasize the importance among adolescent users?

9. As the data will be collected using two languages (i.e., English and French), I wonder if the authors will make sure the measurement invariance or check the linguistic validity between the two languages.

10. The authors may consider using the ASSIST-11, a shortened ASSIST to assess other psychoactive substance use. The ASSIST-11 has been validated across 42 countries recently.

Lee, C. T., Lin, C. Y., Koós, M., Nagy, L., Kraus, S. W., Demetrovics, Z., Potenza, M. N., Ballester-Arnal, R., Batthyány, D., Bergeron, S., Billieux, J., Burkauskas, J., Cárdenas-López, G., Carvalho, J., Castro-Calvo, J., Chen, L., Ciocca, G., Corazza, O., Csako, R. I., Fernandez, D. P., … Bőthe, B. (2023). The eleven-item Alcohol, Smoking and Substance Involvement Screening Test (ASSIST-11): Cross-cultural psychometric evaluation across 42 countries. Journal of Psychiatric Research, 165, 16–27. https://doi.org/10.1016/j.jpsychires.2023.06.033

11. In the Phase 2 project, I would suggest the authors clearly mention the inclusion and exclusion criteria. Current presentation is not that clear. Also, I wonder if the authors will exclude those who have received treatments for cannabis use or internet use. Or, will the authors exclude those who have severe mental health disorder (e.g., schizophrenia)?

12. Lastly, I may overlook. However, the authors did not mention how and where they will deposit the data.

Reviewer #2: This study protocol has been well thought through and very well written/presented. This study is funded by Quebec government and study findings can inform the stakeholders in making public policies regarding adults with problematic Internet use who are also cannabis users. As the authors pointed out - this is one of a kind of the study involving Internet problematic usage and cannabis use, particularly in adult population. The strengths and limitations of this study are acknowledged by the authors adequately. The most challenging aspect of this study is following the mixed method that they are using in obtaining the quantitative and qualitative data, integrating the data and concluding from the qualitative data.

It'll be interesting to see the data, results and conclusions. I recommend for this manuscript to be published with no revisions.

7. PLOS authors have the option to publish the peer review history of their article (what does this mean?). If published, this will include your full peer review and any attached files.

Reviewer #1: No

Reviewer #2: **Yes: **Tulasi Srinivasa Kumar Goriparthi M.D

---

## [Author Response · Author response to Decision Letter 0]

22 Mar 2024

• The authors said, “Problematic Internet use has been associated with cannabis use disorder, especially among adolescent users” in the Abstract. However, the present protocol excludes adolescents. Then, why do the authors want to emphasize the importance among adolescent users?

Thank you for pointing this out. Our intention with this statement was not to emphasize the adolescent population but rather to point out there’s not a lot of evidence of this association other than in research on adolescent populations. The phrasing has been adjusted to express the lack of studies in the adult population instead. 

• [R1] ll74-75, the statement “The COVID-19 pandemic has made problematic Internet use a priority issue for many…” needs more citation in addition to the current consensus statement. The following references may work.

Alimoradi, Z., Lotfi, A., Lin, C.-Y., Griffiths, M. D., & Pakpour, A. H. (2022). Estimation of behavioral addiction prevalence during COVID-19 pandemic: A systematic review and meta-analysis. Current Addiction Reports, 9, 486-517.

Ruckwongpatr, K., Paratthakonkun, C., Ghavifekr, S., Gan, W. Y., Tung, S. E. H., Nurmala, I., Nadhiroh, S. R., Pramukti, I., & Lin, C.-Y. (2022). Problematic Internet Use (PIU) in Youth: A Brief Literature Review of Selected Topics. Current Opinion in Behavioral Sciences, 46, 101150.

Chen, I.-H., Chen, C.-Y., Liu, C.-h., Ahorsu, D. K., Griffiths, M. D., Chen, Y.-P., Kuo, Y.-J., Lin, C.-Y., Pakpour, A. H., Wang, S.-M. (2021). Internet addiction and psychological distress among Chinese schoolchildren before and during the COVID-19 outbreak: A latent class analysis. Journal of Behavioral Addictions, 10(3), 731-746.

Thank you for these suggestions. After examining the proposed references, we decided to cite the reference by Alimoradi et al. (2022) as it gathers a rich body of literature that supports our claim. The other two proposed references have been left out given they are either not specific to the COVID-19 pandemic context (Ruckwongpatr et al. (2022)) or focus exclusively on children (Chen et al. (2021)), while our study is interested in the adult population only. 

• [R1] Line 80, I think that (10,13,14) should be [10,13,14].

This change has been made. Thank you.

• [R1] When the authors propose their first objective (i.e., identify factors associated with problematic Internet use among cannabis users), it is unclear to me if this objective is based on hypotheses or is exploratory. This is important because if the objective will be investigated using hypotheses, the authors should provide theoretical background to support. However, if it is exploratory, the authors do not need to provide theoretical background. But even it is exploratory, the authors should provide some directions regarding what factors are of interests.

Thank you for pointing this out. The objectives of our research remain exploratory. We have added a phrase to the first objective to provide more context as to what factors are of interest to us. 

• [R1] The authors want to recruit 1500 participants for the first phase of the present project, and the proposed sample size calculation is vague. Specifically, the authors simply said that “Sample size (n = 1500) was determined according to the number of participating individuals needed for the development of a regression model, composed of 10 variables associated with the epidemiological triad "host-agent-environment" [17] to determine the relationship between cannabis use and Internet use”. How exactly the sample size is calculated remains unclear to me. 

More information regarding how sample size was determined for this study has been added in lines 130 through 134.

• [R1] Following the previous comment, I think that “sample size calculation” deserves to be a subsection instead of mentioning in the Data analysis section.

Thank you for this suggestion. A subsection has been created for “sample size calculation” after the section “Recruitment and sampling.”

• [R1] It is unclear to me how the authors can recruit the participants to be representative in terms of age, sex and gender, language, and geographic region. Say, what are the proposed sample size for each stratified group? 

A phrase has been added to the ‘Recruitment and sampling’ section to better explain how the stratified random sampling is applied to ensure representativeness. 

• [R1] Also, it is unclear how the representativeness of the database in the firm which will help distribute the questionnaire.

A phrase has been added in the ‘Recruitment and sampling’ section stating that the web panels firm in charge of the recruitment has a respondent pool of over 200,000 members in Quebec, to which stratified random sampling is applied to ensure representativeness of this population.

• [R1] As the data will be collected using two languages (i.e., English and French), I wonder if the authors will make sure the measurement invariance or check the linguistic validity between the two languages.

It has been specified on lines 145 and 203-204 that validated questionnaires used for our data collection are available in both English and French, with the corresponding references provided. 

• [R1] The authors may consider using the ASSIST-11, a shortened ASSIST to assess other psychoactive substance use. The ASSIST-11 has been validated across 42 countries recently.

Lee, C. T., Lin, C. Y., Koós, M., Nagy, L., Kraus, S. W., Demetrovics, Z., Potenza, M. N., Ballester-Arnal, R., Batthyány, D., Bergeron, S., Billieux, J., Burkauskas, J., Cárdenas-López, G., Carvalho, J., Castro-Calvo, J., Chen, L., Ciocca, G., Corazza, O., Csako, R. I., Fernandez, D. P., … Bőthe, B. (2023). The eleven-item Alcohol, Smoking and Substance Involvement Screening Test (ASSIST-11): Cross-cultural psychometric evaluation across 42 countries. Journal of Psychiatric Research, 165, 16–27. https://doi.org/10.1016/j.jpsychires.2023.06.033

Thank you for this suggestion, we will consider it for future projects. However, the protocol approved by the Research Ethics Committee and the institution providing funding for this research uses the 8-item ASSIST scale.

• [R1] In the Phase 2 project, I would suggest the authors clearly mention the inclusion and exclusion criteria. Current presentation is not that clear. Also, I wonder if the authors will exclude those who have received treatments for cannabis use or internet use. Or, will the authors exclude those who have severe mental health disorder (e.g., schizophrenia)?

Thank you for your comment. Phase 2 is a qualitative phase that aims to explore the experience of people with problematic Internet use (including those having sought treatment and/or diagnosed with a mental health disorder). We will therefore not exclude them from the interviews as we wish to study a plurality of experiences. To make the recruitment and sampling section clearer, for Phase 2, inclusion criteria has been numbered as such: “1) have participated in Phase 1 of the study, 2) have agreed to be contacted for Phase 2, and 3) present a problematic use of the Internet, according to IAT score during Phase 1, i.e., a score greater than 50”.

• [R1] Lastly, I may overlook. However, the authors did not mention how and where they will deposit the data.

Data will be stored on the Université de Sherbrooke's secure server and will be accessible upon reasonable request. 

A statement regarding data availability has been added to the article after the Acknowledgements section.

---

## [Decision Letter · Decision Letter 1]

3 May 2024

PONE-D-23-22618R1Cannabis use, mental health, and problematic Internet use in Quebec: A Study ProtocolPLOS ONE

Dear Dr. Brodeur,

Thank you for submitting your manuscript to PLOS ONE. After careful consideration, we feel that it has merit but does not fully meet PLOS ONE’s publication criteria as it currently stands. Therefore, we invite you to submit a revised version of the manuscript that addresses the points raised during the review process.

**ACADEMIC EDITOR: **Dear Authors, The reviewers have responded positively with feedback about your paper. There is one comment from one reviewer that encourages you to cite another article. Please do not feel the need to do so. You can feel free to just respond to the reviewers comment and make any changes that you see fit. Following your response to reviewer we will proceed with decisions. Again, to reiterate, any recommendations by reviewers to cite an article is optional and at discretion of the authors.  Wishing you the best

We look forward to receiving your revised manuscript.

Kind regards,

Souparno Mitra, M.D.

Academic Editor

PLOS ONE

Journal Requirements:

Reviewers' comments:

Reviewer's Responses to Questions

**Comments to the Author**

1. Does the manuscript provide a valid rationale for the proposed study, with clearly identified and justified research questions?

Reviewer #1: Yes

Reviewer #3: Yes

2. Is the protocol technically sound and planned in a manner that will lead to a meaningful outcome and allow testing the stated hypotheses?

Reviewer #1: Yes

Reviewer #3: Yes

3. Is the methodology feasible and described in sufficient detail to allow the work to be replicable?

Reviewer #1: Yes

Reviewer #3: Yes

4. Have the authors described where all data underlying the findings will be made available when the study is complete?

Reviewer #1: Yes

Reviewer #3: Yes

5. Is the manuscript presented in an intelligible fashion and written in standard English?

Reviewer #1: Yes

Reviewer #3: Yes

6. Review Comments to the Author

You may also provide optional suggestions and comments to authors that they might find helpful in planning their study.

Reviewer #1: The authors have nicely responded to my previous comments and made great improvement. I only have one more minor suggestion. That is, the authors have cited Alimoradi et al.'s meta-analysis to support this statement, "The COVID-19 pandemic has made problematic Internet use a priority issue for many entities both in Quebec and abroad (government, researchers, health professionals, etc.), as it has led to increased use of the Internet and new technologies". I would suggest they further indicate that such internet use is associated with poor mental health as shown in another recent meta-analysis paper authored by the same team of Alimoradi.

Ref: Alimoradi, Z., Broström, A., Potenza, M. N., Lin, C.-Y., & Pakpour, A. H. (2024). Associations between behavioral addictions and mental health concerns during the COVID-19 pandemic: A systematic review and meta-analysis. Current Addiction Reports. https://doi.org/10.1007/s40429-024-00555-1

Reviewer #3: Overall, this is a clear, and well-written manuscript. The introduction is relevant. This is an interesting study and the data will be informative. The study design is interesting. The manuscript is structured. It is appropriate in length.

7. PLOS authors have the option to publish the peer review history of their article (what does this mean?). If published, this will include your full peer review and any attached files.

Reviewer #1: No

Reviewer #3: No

---

## [Author Response · Author response to Decision Letter 1]

7 May 2024

You will find below our response to the point raised by Reviewer 1 [R1].

Introduction

• [R1] The authors have nicely responded to my previous comments and made great improvement. I only have one more minor suggestion. That is, the authors have cited Alimoradi et al.'s meta-analysis to support this statement, "The COVID-19 pandemic has made problematic Internet use a priority issue for many entities both in Quebec and abroad (government, researchers, health professionals, etc.), as it has led to increased use of the Internet and new technologies". I would suggest they further indicate that such internet use is associated with poor mental health as shown in another recent meta-analysis paper authored by the same team of Alimoradi.

Ref: Alimoradi, Z., Broström, A., Potenza, M. N., Lin, C.-Y., & Pakpour, A. H. (2024). Associations between behavioral addictions and mental health concerns during the COVID-19 pandemic: A systematic review and meta-analysis. Current Addiction Reports. https://doi.org/10.1007/s40429-024-00555-1

Response: Thank you for this suggestion. We have included a phrase in the introduction indicating the association between problematic Internet use and poor mental health, and its corresponding citation. Accordingly, the source has been added to the reference list.

---

## [Decision Letter · Decision Letter 2]

17 May 2024

Cannabis use, mental health, and problematic Internet use in Quebec: A Study Protocol

PONE-D-23-22618R2

Dear Dr. Brodeur,

We’re pleased to inform you that your manuscript has been judged scientifically suitable for publication and will be formally accepted for publication once it meets all outstanding technical requirements.

Kind regards,

Souparno Mitra, M.D.

Academic Editor

PLOS ONE

Additional Editor Comments (optional):

Reviewers' comments:

Reviewer's Responses to Questions

**Comments to the Author**

1. Does the manuscript provide a valid rationale for the proposed study, with clearly identified and justified research questions?

Reviewer #1: Yes

Reviewer #3: Yes

2. Is the protocol technically sound and planned in a manner that will lead to a meaningful outcome and allow testing the stated hypotheses?

Reviewer #1: Yes

Reviewer #3: Yes

3. Is the methodology feasible and described in sufficient detail to allow the work to be replicable?

Reviewer #1: Yes

Reviewer #3: Yes

4. Have the authors described where all data underlying the findings will be made available when the study is complete?

Reviewer #1: Yes

Reviewer #3: Yes

5. Is the manuscript presented in an intelligible fashion and written in standard English?

Reviewer #1: Yes

Reviewer #3: Yes

6. Review Comments to the Author

You may also provide optional suggestions and comments to authors that they might find helpful in planning their study.

Reviewer #1: The authors have satisfactorily revised the manuscript with the citation of the latest evidence in this field. The protocol is now very clear and ready for publication.

Reviewer #3: Overall, this is a clear, and well-written manuscript. The introduction is relevant. This is an interesting study and the data will be informative. The study design is interesting. The manuscript is structured. It is appropriate in length.

7. PLOS authors have the option to publish the peer review history of their article (what does this mean?). If published, this will include your full peer review and any attached files.

Reviewer #1: No

Reviewer #3: No

---

## [Editor Report · Acceptance letter]

24 May 2024

PONE-D-23-22618R2 

PLOS ONE

Dear Dr. Brodeur, 

I'm pleased to inform you that your manuscript has been deemed suitable for publication in PLOS ONE. Congratulations! Your manuscript is now being handed over to our production team.

Kind regards, 

on behalf of

Dr. Souparno Mitra 

Academic Editor

PLOS ONE